# SIMULATING TASK-FREE CONTINUAL LEARNING STREAMS FROM EXISTING DATASETS

## ABSTRACT

Task-free continual learning is the subfield of machine learning that focuses on learning online from a stream whose distribution changes continuously over time. However, previous works evaluate task-free continual learning using streams with distributions that change only at a few distinct points in time. In order to address the discrepancy between the definition and evaluation of task-free continual learning, we propose a principled algorithm that can permute any labeled dataset into a stream that is continuously nonstationary. We empirically show that the streams generated by our algorithm are less structured than the ones conventionally used in the literature. Moreover, we use our simulated task-free streams to benchmark multiple methods applicable to the task-free setting. We hope that our work will make it more likely that task-free continual learning methods are able to better generalize to real-world problems.

## 1 INTRODUCTION

The dominant paradigm in the field of machine learning involves building a model using a static set of pre-collected data (Mitchell, 1997; LeCun et al., 2015). Unfortunately, it might not be always possible to stick to this paradigm. For instance, animals and humans extract knowledge from their observations continually, and under changing circumstances (Parisi et al., 2019). The field of *continual learning* studies exactly this problem—namely, how to train a machine learning model using data provided by a nonstationary distribution (Aljundi et al., 2019c; Chrysakis & Moens, 2020).

Within the continual learning literature, different underlying assumptions give rise to a number of distinct continual learning settings. Such assumptions might be about whether the data distribution is continuously nonstationary or not, or about whether the model optimization takes place online (with small minibatches of data) or offline (with large batches of data) (De Lange et al., 2021). In this paper, we focus on *task-free* continual learning, which we consider to be the setting closest to how humans and animals learn. In task-free continual learning, the data distribution is assumed to be continuously nonstationary and the optimization takes place online (Aljundi et al., 2019b).

The observation that motivated this work is that there is a large gap between how task-free continual learning is defined and how it is evaluated. In fact, previous works evaluate task-free continual learning using streams with data distributions that are not continuously nonstationary, but change only at a few distinct moments in time and remain stationary otherwise (Aljundi et al., 2019b; Jin et al., 2021). With this work, we aim to bridge the gap between the definition and the evaluation of task-free continual learning.

Our contributions are the following. First, we provide a principled algorithm that can reorder any labeled dataset into a simulated task-free (STF) continual learning stream. This algorithm was designed with the goal of introducing as little design bias as possible to the streams it constructs. Second, we perform a detailed comparison between STF streams generated by the proposed algorithm and the type of streams conventionally used in previous works. Via this comparison, we detail a number of different ways the streams conventionally used are different to our STF streams. Third, we transform four well-known datasets into STF streams, and use them to benchmark a number of methods applicable to task-free continual learning.

The remainder of the paper is structured as follows. In Section 2, we provide an introduction to continual learning and online continual learning, and extensively discuss the gap between the definition

and evaluation of task-free continual learning. In Section 3, we present our algorithm for generating STF streams, and motivate its design. In Section 4, we present and discuss our experiments, and, finally, in Section 5, we summarize our work, discuss its limitations, and offer a future perspective.

## 2 BACKGROUND

### 2.1 CONTINUAL LEARNING

In general, *continual learning* is defined as learning from data that are generated by a *nonstationary* distribution, that is to say, a distribution that changes over time (Zenke et al., 2017; Kurle et al., 2019; Chrysakis & Moens, 2020). An alternative definition of continual learning is the learning of a sequence of tasks over time (Van de Ven & Tolias, 2019; Prabhu et al., 2020; De Lange et al., 2021). But, what is a task?

In the context of continual learning, the term *task* is generally used to describe a collection of data which the model observes in an independent and identically distributed (iid) manner. Tasks are often assumed to be class-disjoint, that is, if data from a particular class appear in a task, no data from the same class will be present in any other task (Van de Ven & Tolias, 2019; Prabhu et al., 2020). Previous works sometimes assume access to *task labels*, which explicitly inform the learner to which task each data point belongs (Nguyen et al., 2018; Zenke et al., 2017). The setting of *class-incremental* continual learning assumes that task labels are only given during training, while the setting of *task-incremental* continual learning assumes access to task labels both during training and during evaluation (Van de Ven & Tolias, 2019; De Lange et al., 2021; Lomonaco & Rish, 2021).

Besides access to task labels, another distinction can be made with regard to whether continual learning takes place online or offline. In the *offline* setting, the learner has access to all data from the present task and can perform multiple passes over these data (De Lange et al., 2021; Prabhu et al., 2020). Conversely, in the *online* setting, the learner receives data from a nonstationary stream in the form of small minibatches, and only has access to one of those minibatches at a time (Aljundi et al., 2019c; Chrysakis & Moens, 2020; Cai et al., 2021).

### 2.2 ONLINE CONTINUAL LEARNING SETTINGS

To avoid potential confusion, we offer precise definitions for online, task-agnostic, and task-free continual learning. First, online continual learning has evolved[1] to be an umbrella term that encompasses all settings in which a model should be trained online using small minibatches of data that are generated by a nonstationary stream (Aljundi et al., 2019a; Pham et al., 2020; Yin et al., 2021).

Task-agnostic and task-free continual learning are both types of online continual learning. In *task-agnostic* continual learning the stream is assumed to be a sequence of tasks but without task labels being available. In other words, the stream consists of a number contiguous iid sub-streams (each one corresponding to a task), and the distribution only changes when there is a transition from one sub-stream to the next. In this setting, however, it is relatively easy to infer task labels during training (Zeno et al., 2018; Kirichenko et al., 2021).

Finally, in *task-free* continual learning the concept of a data distribution that changes at distinct points during learning, is generalized to one that changes constantly over time (Aljundi et al., 2019b). Therefore, in a task-free stream, there are no iid sub-streams, hence the concepts of tasks, task labels, and task boundaries cannot be defined.

### 2.3 TASK-FREE CONTINUAL LEARNING AND ITS EVALUATION

We argue that, in terms of its applicability, task-free continual learning is the most general continual learning setting. To understand why, we need to consider the various aforementioned settings in the context of the simplifying assumptions they make. The most widely adopted assumptions are a) the existence of tasks, b) task labels during training, c) concurrent access to all data from the present

---

[1]We write *evolved* because online continual learning was originally defined to be a nonstationary online learning problem without access to task labels (Aljundi et al., 2019c). A number of subsequent works, however, study online continual learning and do assume access to task labels (Pham et al., 2020; Yin et al., 2021).

task, and d) task labels during evaluation. Generally speaking, the more simplifying assumptions a setting adopts, the more niche this setting is, but also, the less applicable in real-life situations it becomes. The task-incremental setting assumes all four, the class-incremental setting assumes the first three, and the task-agnostic setting, in theory, assumes only the first (but, as we discussed earlier, task labels during training can be inferred). In the task-free setting, however, there are no simplifying assumptions. Put another way, task-free continual learning adopts the most general definition of continual learning.

To reinforce this point, let us consider the four real-life continual learning scenarios identified by Farquhar & Gal (2018): a) a disease-diagnosis system trained incrementally with data different from different populations; b) a wind-turbine-safety system that learns to predict when to deactivate the turbine in order to prevent damage from strong winds; c) a recommender system that learns to serve ads tailored to a user's needs and interests; d) a exploration rover that learns to navigate the various terrains of the planet Mars. These examples were meant to describe continual learning in general, but, interestingly, three out of four are task-free continual learning problems (the only exception is the first one). Indeed, the latter three problems all involve data distributions that change continuously over time (seasonal and climate changes, changes in trends and individual interests, and terrain changes, respectively), not in distinct steps. If these examples are any indication, many real-life continual learning problems are task-free.

Unfortunately, there is a significant discrepancy between how task-free continual learning is defined and how it is evaluated. Due to the lack of appropriate task-free benchmarks, previous works proposing methods that do not make assumptions about the nature of the input stream, evaluate their performance on streams that are not continuously nonstationary (Aljundi et al., 2019a; Jin et al., 2021). Therefore, we argue that task-free continual learning should be evaluated using task-free streams. One way to achieve this goal, would be to build new *ordered* datasets from real-world task-free continual learning problems. However, this process is slow and, potentially, very expensive. Instead, we developed a principled algorithm that can transpose any labeled dataset into a stream that is continuously nonstationary. We describe and motivate this algorithm in the following section.

## 3 METHODOLOGY

### 3.1 PROBLEM FORMULATION

Let $\mathbf{D} = \{(\boldsymbol{x}_i, y_i)\}_{i=1}^{n}$ be an arbitrary labeled dataset of size $n$, where $\boldsymbol{x}_i$ are the data instances and $y_i$ are their corresponding labels. This dataset contains data instances $\boldsymbol{x}_i$ of $c$ distinct classes, that is to say, for $i = 1, \ldots, n$, it is $y_i \in \{1, \ldots, c\}$.

Our goal is to permute the order in which the data instances appear within the dataset, so that when the permuted dataset is broken down in small mini-batches, it approximates the characteristics of task-free continual learning streams. In intuitive terms, we want the data distribution of the resulting streams to be changing throughout the duration of learning, and not just at distinct points in time (as is the case in non-task-free settings). Moreover, we want the resulting streams to contain as little design bias as possible, in order for them to serve as benchmarks that generalize adequately to real-world task-free continual learning problems. Since such streams are an attempt to simulate the characteristics of real-world task-free continual learning streams, we will call them simulated task-free (STF) streams.

Formally, our goal is to assign to each data instance $\boldsymbol{x}_i$ a permutation index $p_i$ that specifies in which position $\boldsymbol{x}_i$ will appear in the permutation. In particular, if $p_i = k$ the data instance $\boldsymbol{x}_i$, which was the $i$-th instance in the dataset's original order, will appear as the $k$-th instance in the permuted order.

We break this problem down into two sub-problems. First, in Section 3.2, we discuss how to assign to each class $j$ a one-dimensional distribution $\mathcal{D}_j$ (for all $j = 1, \ldots, c$). Then, in Section 3.3, we explain how to use the assigned distributions $\mathcal{D}_j$ to generate a dataset permutation.

### 3.2 ASSIGNING A DISTRIBUTION TO EACH CLASS

Let $t \in [0, 1]$ be the time during which the continual learning takes place, where we assume that learning starts at $t = 0$ and ends at $t = 1$. We define the class distributions $\mathcal{D}_j$ as distributions over

the random variable $t$. At a high level, the time distribution $\mathcal{D}_j(t)$ of class $j$ will determine how early or late in the stream instances of class $j$ are likely to appear compared to instances of the other classes. For instance, if $\mathbb{E}\big[\mathcal{D}_1(t)\big] > \mathbb{E}\big[\mathcal{D}_2(t)\big]$, that is, the mean of the time distribution of class 1 is greater than the mean of the time distribution of class 2, then instances of class 1 are more likely to appear in the stream later than those of class 2. In addition, the standard deviation of the time distribution of each class will determine whether its instances are likely to appear more concentrated or more dispersed over the stream.

Given the information in the previous paragraph, there are several questions that need to be answered. We start by describing the principle of maximum entropy (Jaynes, 1957a;b), and how we apply it in order to assign a mean $\mu_j$ and a standard deviation $\sigma_j$ to each class $j$. Subsequently, we discuss and motivate which family of distributions we decided to use. Finally, we explain how to derive the parameters of each class's distribution given its mean $\mu_j$ and its standard deviation $\sigma_j$.

The principle of maximum entropy states that when selecting what kind of distribution to use to represent current knowledge about a system, out of all the distributions consistent with this current knowledge, one should select the distribution with the maximum entropy

---

**Algorithm 1** Assign a distribution to each class.

 Number of classes $c$
 Desired average standard deviation $\mu_\sigma$

1: Find rate $\lambda$ such that $\frac{\gamma}{1-e^{-\lambda\gamma}} - \frac{1}{\lambda} = \mu_\sigma$
2: **for** class $j$ in $1, \ldots, c$ **do**
3:     Sample the standard deviation: $\sigma_j \sim \mathcal{E}(\sigma \mid \lambda, \gamma)$
4:     Compute $r_j = \sqrt{\frac{1}{4} - \sigma_j^2}$
5:     Sample the mean: $\mu_j \sim \mathcal{U}\big(\mu \mid 0.5 - r_j, 0.5 + r_j\big)$
6:     Compute $\alpha_j = \mu_j \left[ \frac{\mu_j(1-\mu_j)}{\sigma_j^2} - 1 \right]$
7:     Compute $\beta_j = (1 - \mu_j) \left[ \frac{\mu_j(1-\mu_j)}{\sigma_j^2} - 1 \right]$
8:     Set $\mathcal{D}_j = \mathcal{B}(\alpha_j, \beta_j)$
9: **end for**

---

(Jaynes, 1957a;b). Intuitively, the maximum-entropy distribution is the most uninformative distribution consistent with current knowledge. Hence, by choosing the maximum-entropy distribution, the user takes into account only what the current knowledge suggests, without adding any unnecessary bias (Jaynes, 1968).

In order to use the maximum-entropy principle to sample the means $\mu_j$, we need to first consider what our current knowledge about them is. Since the class distributions $\mathcal{D}_j(t)$ are defined on the interval $[0, 1]$, their corresponding means $\mu_j$ should also be contained in the same interval. Hence we are looking for the maximum-entropy distribution defined over the closed interval $[0, 1]$. This distribution is the uniform (Udwadia, 1989):

$$\mathcal{U}(\mu \mid 0, 1) = \begin{cases} 1, & \text{for } \mu \in [0, 1] \\ 0, & \text{elsewhere.} \end{cases} \tag{1}$$

Now we move on to sampling the standard deviations $\sigma_j$. Once again, we need to consider what our current knowledge suggests. Since the class distributions $\mathcal{D}_j(t)$ are defined on the interval $[0, 1]$, it must hold that $0 \le \sigma_j \le 0.5$ for all $j$.[2] Also, in contrast to how we sample the means, here we would like to be able to manually set the average standard deviation $\mu_\sigma$ over all classes (in Section 4.4 we show that by changing the value of $\mu_\sigma$, the resulting streams can become easier or harder to learn from). In short, we are looking for the maximum-entropy distribution that is defined on the interval $\sigma \in [0, 0.5]$, and of which the mean value is $\mu_\sigma$. This distribution is the truncated-exponential (Udwadia, 1989), and is defined as

$$\mathcal{E}(\sigma \mid \lambda, \gamma) = c e^{\lambda\sigma}, \quad \sigma \in [0, \gamma], \tag{2}$$

where $c$ is the normalizing constant, and $\gamma$ is the truncation parameter, which in our case is set to 0.5. The parameter $\lambda$ is called the rate of the distribution and is set so that the expected value of the truncated exponential is equal to the desired value $\mu_\sigma$:

$$\mathbb{E}[\sigma] = \frac{\gamma}{1 - e^{-\lambda\gamma}} - \frac{1}{\lambda} = \mu_\sigma. \tag{3}$$

---

[2]This result follows directly from Popoviciu's inequality on variances (Popoviciu, 1935).

We discuss the truncated exponential more extensively in the appendix (including how to compute the normalization constant, how to find the appropriate rate $\lambda$ given the desired mean $\mu_\sigma$, and how to draw samples from it).

For deciding which distribution family to use, we could once again try to make use of the maximum entropy principle. We have sampled a mean $\mu_j$ and a standard deviation $\sigma_j$ for each class $j$ in our dataset, and now we want to assign to that class a distribution defined on the interval $[0, 1]$, with the same mean and standard deviation. The maximum-entropy distribution of a specified mean and standard deviation, and also defined on a bounded interval is called the truncated normal (Udwadia, 1989). However, we would also like to be able to easily derive the parameters of each distribution given its mean and standard deviation. In the case of the truncated normal, deriving its parameterization involves solving a non-linear system of equations, which does not have an analytical solution and is not guaranteed to be solvable in a numerically stable way. Instead, we argue that the Beta distribution is a more appropriate choice for a number reasons. First, deriving the parameters of a Beta given a desired mean and standard deviation is trivial (see Eq. 5). Second, as we empirically show in the appendix, the Beta captures $99.84\%$ of the entropy of the truncated normal, on average. Third, the Beta is mathematically convenient for our use-case since its support is the interval $[0, 1]$.

The Beta distribution is defined as

$$\mathcal{B}(\alpha, \beta) = cx^{\alpha-1}(1-x)^{\beta-1}, \quad x \in [0, 1], \tag{4}$$

and is parameterized by its shape parameters $\alpha$ and $\beta$, while $c$ is a normalization constant. Given a desired mean $\mu_j$ and standard deviation $\sigma_j$, the shape parameters $\alpha_j$ and $\beta_j$ of a Beta with such a mean and a standard deviation are computed as follows:

$$\alpha_j = \mu_j \left[ \frac{\mu_j(1-\mu_j)}{\sigma_j^2} - 1 \right], \quad \beta_j = (1-\mu_j) \left[ \frac{\mu_j(1-\mu_j)}{\sigma_j^2} - 1 \right]. \tag{5}$$

However, we need to be able to guarantee the existence of a distribution with support $[0, 1]$ given the mean $\mu_j$ and the standard deviation $\sigma_j$ that we have sampled for each class $j$. With regard to the Beta distribution, the relevant necessary condition is

$$\sigma_j^2 < \mu_j(1-\mu_j). \tag{6}$$

Therefore, we need to make sure this condition holds for every class $j$. A simple way to ensure that, would be to first sample a mean $\mu_j$ in $[0, 1]$ as described above, and then to sample the standard deviation $\sigma_j$, with rejection sampling (Casella et al., 2004), until we find a pair $(\mu_j, \sigma_j)$ that satisfies Eq. 6. An alternative would be to first sample the standard deviation $\sigma_j$, and then to shrink the support of the uniform distribution from which $\mu_j$ is sampled, in order to guarantee that Eq. 6 will be satisfied for any choice within the shrunk support. After some algebra, we get the shrunk support:

$$\left[ 0.5 - r_j, 0.5 + r_j \right], \quad \text{where } r_j \triangleq \sqrt{\frac{1}{4} - \sigma_j^2}. \tag{7}$$

In our view, using the shrunk-support approach is superior to rejection sampling since it does not require repeated sampling steps to succeed. The entire sampling process is presented in Algorithm 1.

### 3.3  PERMUTING THE DATASET

Now we will describe how to use the class distributions $\mathcal{D}_j = \mathcal{B}(\alpha_j, \beta_j)$ to permute the dataset $\mathbf{D}$ (see also Algorithm 2). First, we assign a *timestamp* $t_i$ to each instance $i$ of the dataset. These timestamps will then be used to produce a permutation $\boldsymbol{p}$, according to which we will permute the dataset $\mathbf{D}$.

For each data instance $(\boldsymbol{x}_i, y_i)$, we sample a timestamp from the distribution of its class. In other words, we set $j = y_i$, and then sample $t_i \sim \mathcal{B}(\alpha_j, \beta_j)$. Hence, we see that the timestamps of all data instances of a particular class $j$ are sampled from the same distribution, namely, $\mathcal{B}(\alpha_j, \beta_j)$. Afterwards, we compute the permutation $\boldsymbol{p} = (p_1, \ldots, p_n)$ as the vector of indexes that would sort the vector $(t_1, \ldots, t_n)$. In other words, $(p_1, \ldots, p_n)$ is computed by applying the argsort operation on the vector $(t_1, \ldots, t_n)$. Finally, we permute the dataset according to $\boldsymbol{p}$. Intuitively, in the permuted dataset, the data instance with the smallest timestamp will appear first, the one with the second-smallest timestamp will appear second, while the one with the largest timestamp will appear last. A toy example of a dataset permutation is presented in Figure 1.

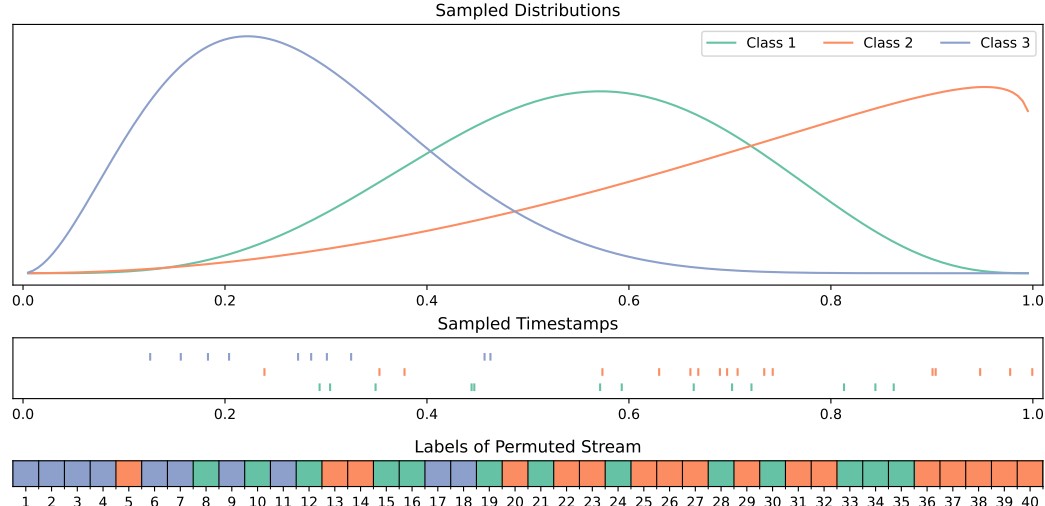

Figure 1: We use a toy dataset with 13, 17, and 10 instances, for class 1, 2, and 3, respectively. A task-free stream is constructed by sorting the timestamps sampled from the class distributions.

## 4 EXPERIMENTS

### 4.1 EXPERIMENTAL SETTINGS

**Datasets** We use four datasets of varying difficulty. EMNIST (Cohen et al., 2017) is a dataset containing approximately 130,000 grayscale images of handwritten characters and digits belonging to 47 classes. CIFAR-10 and CIFAR-100 (Krizhevsky, 2009) are datasets that contain each 50,000 color images of 10 and 100 classes, respectively. Finally, tinyImageNet (Le & Yang, 2015) is the most challenging dataset widely used in evaluating continual learners. It contains 100,000 color images of 200 different classes. We have not used any data augmentation in our experiments, since we want to keep our evaluation domain-agnostic, and data augmentation might not be possible or practical for data modalities other than images.

---

**Algorithm 2** Permute the dataset.

Dataset $\mathbf{D} = \{(\boldsymbol{x}_i, y_i)\}_{i=1}^n$
Class distributions $\mathcal{B}(\alpha_j, \beta_j)$, for $j = 1, \ldots, c$

1: **for** data instance $i$ in $1, \ldots, n$ **do**
2:   Set $j = y_i$
3:   Sample timestamp: $t_i \sim \mathcal{B}(\alpha_j, \beta_j)$
4: **end for**
5: Compute permutation: $\boldsymbol{p} = \mathrm{argsort}(t_1, \ldots, t_n)$
6: Permute dataset $\mathbf{D}$ according to permutation $\boldsymbol{p}$

---

**Methods** Experience replay (ER) (Isele & Cosgun, 2018; Chaudhry et al., 2019) is the most fundamental continual learning baseline. It performs replay from a memory which is populated using reservoir sampling (Vitter, 1985). Maximally-interfered retrieval (MIR) (Aljundi et al., 2019a) is an extension of ER that replays the instances which would experience the largest loss increases if the model were to be updated using only the current mini-batch of observations. Class-balancing reservoir sampling (CBRS) (Chrysakis & Moens, 2020) uses a memory population algorithm that maintains the memory as class-balanced as possible at all times. Greedy sampler and dumb learner (GDUMB) (Prabhu et al., 2020) also uses a class-balancing memory population algorithm and trains the model using only data stored in memory.[3] Gradient-based memory editing (GMED) (Jin et al., 2021) edits the data stored in memory in order to make them more challenging to memorize. Finally, asymmetric cross entropy (ACE) (Caccia et al., 2021) employs a modified loss function that improves continual learning performance by reducing representation drift.

---

[3]The original formulation of GDUMB (Prabhu et al., 2020) is not directly applicable to task-free continual learning since it only trains a model after the stream has been observed in its entirety (Verwimp et al., 2021). Nonetheless, it can be easily extended for use in task-free continual learning (please refer to the appendix).

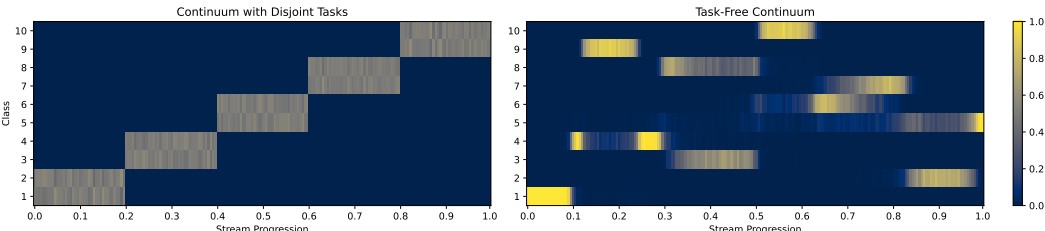

Figure 2: A conventional CIFAR-10 stream with disjoint tasks (left) and a simulated task-free (STF) stream of the same dataset (right). Best viewed zoomed-in and in color.

Table 1: Benchmarking task-free continual learning methods using STF streams of four datasets. We present the final accuracy on the test set after observing the entire stream (Fin. Acc.), and the information retention averaged over the entire stream (Av. IR). All entries are $95\%$-confidence intervals over 20 runs.

|  | EMNIST | | CIFAR-10 | | CIFAR-100 | | tinyImageNet | |
|---|---|---|---|---|---|---|---|---|
|  | Fin. Acc. | Av. IR | Fin. Acc. | Av. IR | Fin. Acc. | Av. IR | Fin. Acc. | Av. IR |
| ER | $80.1 \pm 0.5$ | $87.7 \pm 0.6$ | $38.4 \pm 2.1$ | $56.9 \pm 1.7$ | $14.3 \pm 0.7$ | $27.9 \pm 0.9$ | $8.2 \pm 0.6$ | $16.3 \pm 0.5$ |
| MIR | $79.0 \pm 0.4$ | $88.4 \pm 0.6$ | $39.9 \pm 1.8$ | $56.2 \pm 2.0$ | $14.1 \pm 0.6$ | $29.2 \pm 1.0$ | $7.7 \pm 0.7$ | $16.4 \pm 0.4$ |
| CBRS | $79.7 \pm 0.6$ | $87.0 \pm 0.6$ | $38.1 \pm 2.0$ | $53.2 \pm 1.8$ | $14.4 \pm 0.8$ | $26.6 \pm 0.9$ | $8.6 \pm 0.7$ | $15.7 \pm 0.4$ |
| GDUMB | $81.0 \pm 0.2$ | $88.2 \pm 0.6$ | $41.4 \pm 1.7$ | $53.3 \pm 1.6$ | $12.8 \pm 0.5$ | $23.6 \pm 0.7$ | $6.9 \pm 0.4$ | $14.9 \pm 0.4$ |
| GMED | $80.4 \pm 0.6$ | $88.0 \pm 0.7$ | $39.3 \pm 1.9$ | $56.2 \pm 1.9$ | $14.5 \pm 1.0$ | $28.2 \pm 1.0$ | $8.5 \pm 0.8$ | $16.3 \pm 0.6$ |
| ACE | $80.6 \pm 0.4$ | $89.5 \pm 0.6$ | $49.9 \pm 2.0$ | $64.6 \pm 1.6$ | $19.3 \pm 0.5$ | $32.6 \pm 0.8$ | $11.3 \pm 0.4$ | $20.9 \pm 0.5$ |

**Hyperparameters** Following previous work (Aljundi et al., 2019a; Chrysakis & Moens, 2020; Jin et al., 2021), we use stochastic gradient descent optimization with a learning rate of 0.1, and we set both the stream and replay batch sizes to 10. Method-specific hyperparameters are set based on the values provided in their respective papers. We use memory sizes in the range of 1–4% of the size of the stream (2000 for EMNIST, 1000 for CIFAR-10, 2000 for CIFAR-100, and 4000 for tinyImageNet). Please refer to the appendix for information on the architectures used.

**Evaluation Metrics** Following previous work (Aljundi et al., 2019a; Chrysakis & Moens, 2020; Jin et al., 2021), we evaluate all methods by calculating the accuracy on the unseen testing data after the end of learning. Moreover, in order to also evaluate the longitudinal learning performance of each method throughout the continuum, we use the information retention metric proposed in Cai et al. (2021) (accuracy over past observations) averaged over the entire stream.

### 4.2 STREAM COMPARISON

We start by comparing a conventional CIFAR-10 distinct-task stream with two instances of streams generated by our proposed algorithm (see Figure 2). We split the two streams in 200 chunks, and compute the relative frequency of each class in each of the 200 chunks. The conventional stream (left) is split into 5 tasks with 4 distinct task boundaries between them, and that the data distribution remains stationary within each task. Conversely, in the STF stream (right), the distribution changes continuously over time, sometimes more slowly and others more abruptly. Moreover, we observe other interesting characteristics of the STF stream, such as a) variation in how dispersed or concentrated each class appears over the stream; b) class distributions with more than one modes (e.g., class 4 on the right); and c) class distributions that are skewed (e.g., class 5 on the right). We expect all these characteristics to be present in real-world task-free streams, but, unfortunately, they are never present in the conventional distinct-task streams.

### 4.3 BENCHMARKING

At this point we benchmark six methods applicable to task-free continual learning using our STF streams (see Table 1). We describe in the appendix how we set the average standard deviation $\mu_\sigma$ for each dataset. ER and its variants (MIR, CBRS, GMED) perform similarly across all datasets. GDUMB is different from all the other methods in the sense that it is optimizing a model using

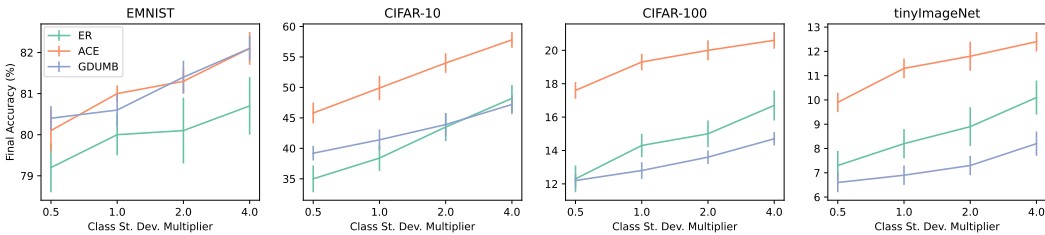

Figure 3: We evaluate the final accuracy of ER, ACE, and GDUMB using STF streams generated from four datasets with the $\mu_\sigma$ values used in Section 4.3 scaled by 0.5, 1, 2, or 4. All results are presented as 95%-confidence intervals.

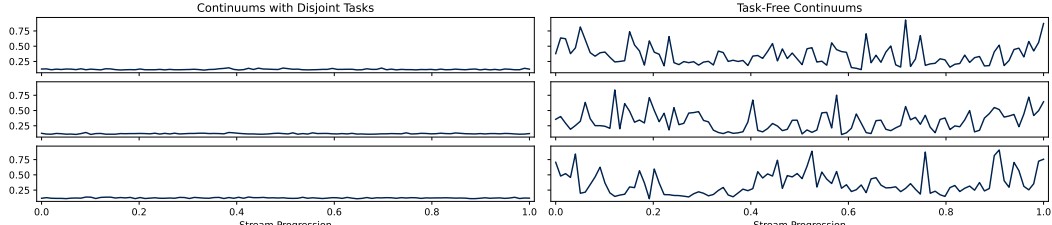

Figure 4: We use CIFAR-100 to create three disjoint-task streams with 10 classes per task (left) and three STF streams (right). For each of the streams, we plot the relative frequency of the most prevalent class at each moment in time.

only data stored in memory. Such an approach appears to be performing well in EMNIST and CIFAR-10, but not so well in the datasets that contain a large number of classes (CIFAR-100, tiny-ImageNet). ACE outperforms all other methods in CIFAR-10, CIFAR-100, and tinyImageNet, a result which suggests that the use of the asymmetric cross-entropy approach can be applied successfully in streams with continuously changing data distributions.

## 4.4 THE EFFECT OF CLASS DISPERSION

Here we examine the effect of the hyperparameter $\mu_\sigma$, which determines how concentrated or spread-out the class distributions are. In Figure 3, we compare ER, ACE, and GDUMB in terms of their final accuracy for four different values of $\mu_\sigma$ (we use the values of $\mu_\sigma$ that we used in Section 4.3 scaled by 0.5, 1, 2, or 4). We observe that for all three methods, streams with more dispersed classes (with larger standard deviations) are easier to learn, and, conversely, streams with more concentrated classes (with smaller standard deviations) are more difficult. Our interpretation of these results is that when the class distributions on average have a higher measure of dispersion, the stream batches are more likely to contain a larger variety of labels, and the model can learn better class-discriminative features. Therefore, we can interpret the value of $\mu_\sigma$ as a kind of difficulty knob for the resulting STF streams. It is also interesting to note what happens in the two extreme cases. When we set $\sigma_j = 0$ for all classes $j$, the resulting streams become equivalent to disjoint-task streams with one class per task. On the other hand, when we set $\sigma_j = 0.5$ for all classes $j$, the resulting streams are iid (or, alternatively, one task that contains all classes).

## 4.5 OTHER CONSIDERATIONS

Finally, we want to note some other ways in which STF streams differ with disjoint-task streams. First, we use the CIFAR-100 dataset to generate three disjoint-task streams and three STF streams. In Figure 4, we plot, for both the disjoint-task (left) and the STF streams (right), the relative frequency of the most prevalent class at each moment in time. Since the disjoint-task streams (left) are constructed with 10 classes per task, the resulting relative frequency is constantly 0.1. On the contrary, the relative frequencies for the STF streams vary in all of the three plots on the right, which is evidence of their lack of structure.

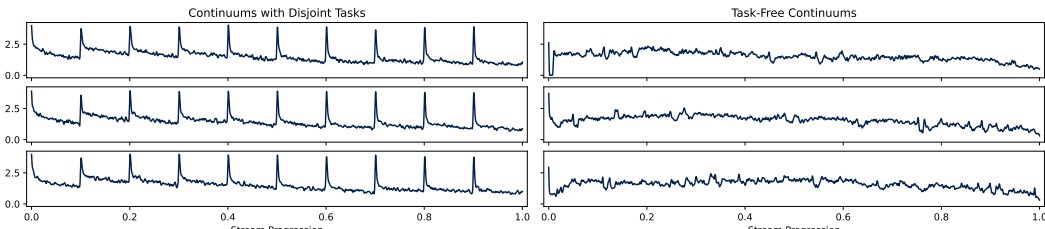

Figure 5: We use CIFAR-100 to create three disjoint-task streams with 10 classes per task (left) and three STF streams (right). We run the ER algorithm on each of the six streams, and plot the corresponding loss curves.

In Figure 5, we plot the loss of the ER method on CIFAR-100, for three disjoint-task streams (left) and three STF streams (right). We observe that, since the disjoint-task streams have always exactly the same structure, their corresponding loss curves are essentially identical (loss spikes take place every time there is a task transition). In contrast, because the STF streams are more varied in terms of their structure, their loss curves are more dissimilar.

## 5 DISCUSSION & CONCLUSION

The general goal of research is to increase our knowledge and our ability to solve complex problems, and task-free continual learning is evidently one of them. Given how generally applicable this problem is (see our arguments in Section 2.3), we believe it is critical to have in place evaluation frameworks that are in line with real-world applications. In the opposite case, we cannot be confident that the algorithms and methodologies that we design will be able to generalize well when applied in the real world. A relevant metaphor would be transfer learning: the closer the source distribution is to the target distribution, the easier it is to transfer knowledge. Applied to our problem, the closer the evaluation framework is to task-free continual learning, the more confident we can be that the methods that perform well in the evaluation framework will also perform well in the real world.

Furthermore, since there is inherent uncertainty in what real-world task-free continual learning streams would be like, we argue that we should not be imposing any unnecessary structure on our evaluation frameworks. As we showed in Section 4.5, however, conventional task-disjoint streams are highly structured. We consider a more general evaluation framework to be more appropriate as a benchmark. Indeed, using on streams with various characteristics in terms of their underlying data distributions (see Figure 2) is a more robust evaluation, than only using streams with exactly the same structure.

One limitation of our stream-simulating algorithm is that it relies on labels, and hence, cannot be readily extended to unlabeled datasets. Future work could examine whether this extension is possible by using unsupervised representations followed by clustering to assign pseudo-labels to each instance of the dataset. Extending our algorithm to multi-label classification problems could be possible by transforming the problem into a multi-class problem (Spolaôr et al., 2013). Moreover, our algorithm could be adapted to regression problems by quantizing the output space.

To summarize, our work is motivated by the observation that the definition and the evaluation of task-free continual learning are not aligned. In particular, task-free continual learning involves data distributions that change continuously over time, but the evaluation of task-free continual learning is performed using data distributions that change only at discrete steps. To help remedy this issue, we have proposed an algorithm that can transform any labeled dataset into a task-free continual learning stream, that is, a stream whose data distribution changes, not just at distinct steps, but continuously over time. We have demonstrated experimentally that the STF streams generated using our algorithm contain much less structure than the disjoint-task streams conventionally used in past work. This lack of structure is, in our view, a desirable feature, since the STF streams can better capture the uncertainty of what a real-world task-free stream would be like. We hope that our work will make it more likely that task-free continual learning contributions proposed in future work will be able to better generalize to practical applications of task-free continual learning.

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

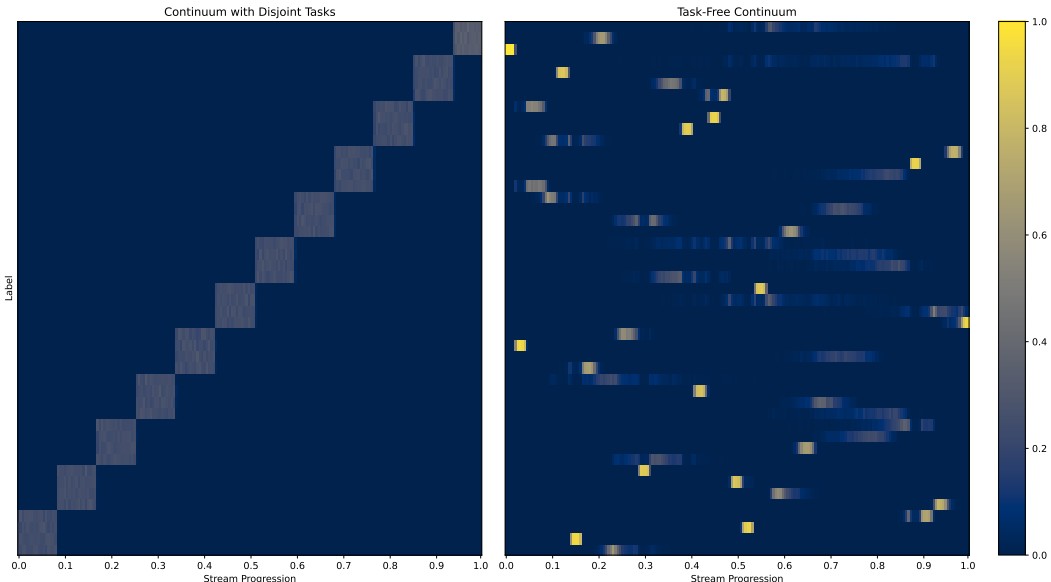

Figure 6: A conventional EMNIST stream with disjoint tasks (left) and a simulated task-free (STF) stream of the same dataset (right). Best viewed zoomed-in and in color.

Table 2: (left) A simple convolutional block; (middle) The Convolutional Neural Network (CNN) architecture used in the EMNIST experiments. (right) The reduced ResNet-18 architecture used for CIFAR-10, CIFAR-100, and tinyImageNet, is built using the BasicBlock($n_f, n_b, n_s$) from (He et al., 2016), where $n_f$ is the number of convolutional filters, $n_b$ is the number of sub-blocks per block, and $n_s$ is the stride of the layer.

| ConvBlock | CNN | Reduced ResNet-18 |
|---|---|---|
| Conv2D($n_{\text{in}}, n_{\text{out}}$) | ConvBlock($1, 32$) | BasicBlock($20, 2, 1$) |
| ReLU | ConvBlock($32, 64$) | BasicBlock($40, 2, 2$) |
| BatchNorm2D($n_{\text{out}}$) | Linear($64, c$) | BasicBlock($80, 2, 2$) |
| Conv2D($n_{\text{out}}, n_{\text{out}}$) | | BasicBlock($160, 2, 2$) |
| ReLU | | AveragePooling |
| BatchNorm2D($n_{\text{out}}$) | | Linear($160, c$) |
| MaxPooling2D($2, 2$) | | |

## A  REPRODUCIBILITY

In order to ensure that our work is not only reproducible but also easily accessible, we commit to open sourcing our code upon acceptance. Moreover, in order to increase the reproducibility of our results, wherever we present average performance over many runs, we do so in the form 95%-confidence intervals, in order for the numbers to accurately reflect their inherent uncertainty.

## B  MORE SIMULATED TASK-FREE STREAMS

In Figure 6, Figure 7, Figure 8, we present a comparison of a conventional distinct-task stream and a simulated task-free (STF) for EMNIST, CIFAR-100, and tinyImageNet respectively.

## C  ARCHITECTURES

Please refer to Table 2

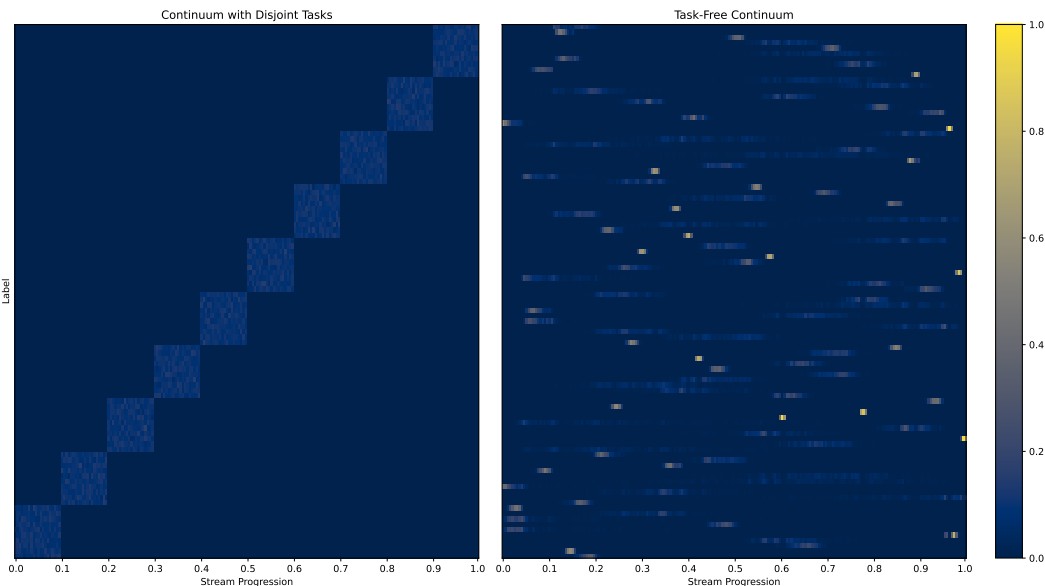

Figure 7: A conventional CIFAR-100 stream with disjoint tasks (left) and a simulated task-free (STF) stream of the same dataset (right). Best viewed zoomed-in and in color.

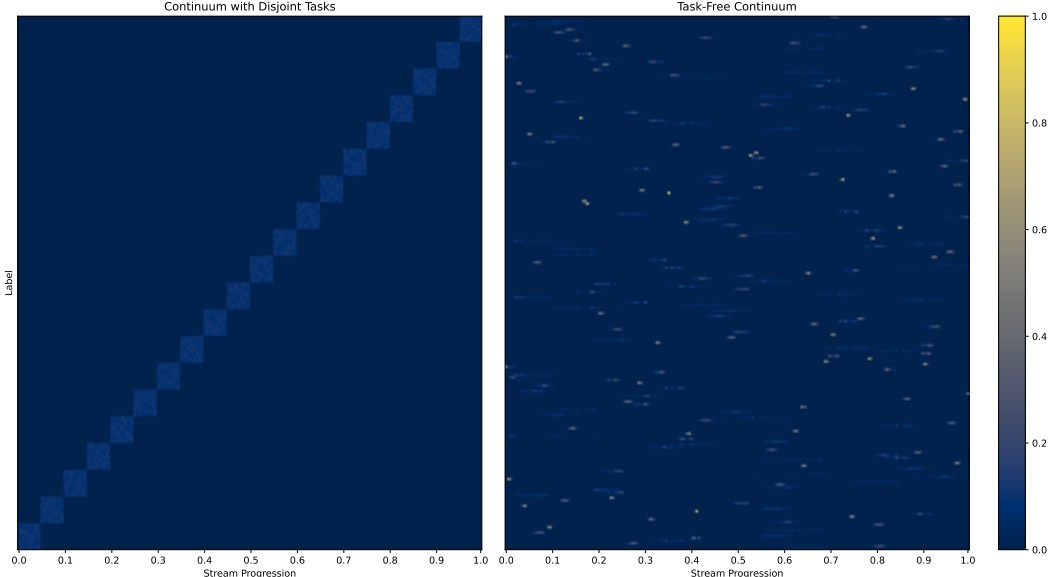

Figure 8: A conventional tinyImageNet stream with disjoint tasks (left) and a simulated task-free (STF) stream of the same dataset (right). Best viewed zoomed-in and in color.

## D    SETTING THE AVERAGE STANDARD DEVIATION

We set the values of $\mu_\sigma$ by associating them to a number of tasks. Previous work typically uses a range of 5–20 number of tasks. In that case, each class's distribution is essentially a uniform with support the inverse of the number of tasks. If the number of tasks is set to $T$, the standard deviation of a uniform with support $1/T$ is $\sqrt{1/12} \times 1/T$. Here, we have used average standard deviations $\mu_\sigma$ which correspond to $T = 12, 5, 10, 20$ tasks for EMNIST, CIFAR-10, CIFAR-100, and tinyImageNet, respectively.

## E    BETA AND THE TRUNCATED NORMAL

Since the truncated normal is the maximum-entropy distribution of a given mean and standard deviation defined on a closed interval, we want to examine how much of its entropy the Beta can capture. We start by sampling a mean $\mu \in [0, 1]$ and a standard deviation $\sigma \in [0, 0.5]$, and we take a normal distribution with these moments and truncate it in $[0, 1]$. We evaluate the mean $\hat{\mu}$ and standard deviation $\hat{\sigma}$ of the truncated normal, and define a Beta with the same moments. Afterwards, we can evaluate the ratio of the entropy of the truncated normal that the Beta can capture. Performing this simulation $10^5$ times, we find that the Beta captures $99.84\%$ of the entropy of the truncated normal. Hence, we argue that the Beta can serve as a substitute for the truncated normal, since, as we mentioned earlier, it is more convenient mathematically.

## F    ON THE TRUNCATED EXPONENTIAL

The *truncated exponential* distribution, with a *rate* parameter $\lambda \neq 0$ and the truncation parameter $\gamma > 0$ for its support, is defined as

$$f_{\mathcal{E}}(x \mid \lambda, \gamma) \triangleq ce^{\lambda x}, \quad x \in [0, \gamma], \tag{8}$$

where $c = \lambda/(e^{\lambda\gamma} - 1)$ is the *normalizing constant*. The mean of a $\mathcal{E}(\lambda, \gamma)$ over its support is computed as

$$\mu = c\int_0^\gamma xe^{\lambda x}dx = \frac{(\lambda\gamma - 1)e^{\lambda\gamma} + 1}{\lambda(e^{\lambda\gamma} - 1)} = \frac{\gamma}{1 - e^{-\lambda\gamma}} - \frac{1}{\lambda}. \tag{9}$$

An appropriate rate can be found by numerically solving the previous equation for specific values of $\mu$ and $\gamma$.

The *cumulative distribution function* (CDF) is

$$F_{\mathcal{E}}(x \mid \lambda, \gamma) = \frac{e^{\lambda x} - 1}{e^{\lambda\gamma} - 1}, \quad x \in [0, \gamma], \tag{10}$$

hence, we can sample from a TE distribution using *inverse-transform* sampling (ITS) as follows:

$$x = \frac{1}{\lambda}\ln\big[(e^{\lambda\gamma} - 1)u + 1\big], \quad \text{where } u \sim \mathrm{U}[0, 1]. \tag{11}$$

