# OpenReview forum: "Simulating Task-Free Continual Learning Streams From Existing Datasets"
_ICLR.cc/2023/Conference — Submitted to ICLR 2023_

### Official Review · Reviewer_A6c5 · 2022-10-22

**Confidence:** 3
**Correctness:** 3
**Technical Novelty And Significance:** 1
**Empirical Novelty And Significance:** 2
**Recommendation:** 3

**Clarity, Quality, Novelty And Reproducibility:**

The writing quality and clarity are decent and the paper can be followed without many obstacles. The algorithmic novelty in the paper is highly limited and the paper does not offer any deeper understanding of the research problem, nor the experiments are informative. The primary contribution is a new evaluation protocol using existing datasets which on its own is not a sufficient contribution for ICLR. The authors have committed to releasing the code after acceptance, however, the codebase is not included as supplementary material. Hence, it is not possible to comment on how reproducible the results are and how good the codebase is documented for the use of the research community. It seems this work will not lead to much impact on the continual learning problem.

**Strength And Weaknesses:**

Strength

1. The writing is clear.

2. Task-free evaluation is a challenge for continual learning and the paper identifies a good research problem.

Weaknesses

1. Novelty is extremely limited.

2. The benchmark is contrived without any connection to a practical application.

3. Experiments are limited and do to convey much understanding because the datasets are on the simple end of continual learning datasets and only a limited number of continual learning algorithms are included.

**Summary Of The Paper:**

The paper develops a pipeline for evaluating task-free continual algorithms using three common machine learning datasets. The core idea is to generate a stream of tasks from a dataset, e.g. mini-Imagenet, and then assign to each data instance a permutation index based on the Beta distribution to determine the time in the stream that it should appear to remove task boundaries. Experiments using four continual learning benchmarks and 6 existing continual learning algorithms are provided in the proposed evaluation protocol.

**Summary Of The Review:**

This paper proposes a benchmark dataset for task-free continual learning and it lacks any algorithmic novelty. The benchmark is also somewhat contrived and is only based on operating on existing datasets. As a result, the only novelty is how to process a dataset such that it can be used for task-free evaluation and there is no real connection to a practical application. The paper is also more related to a conference like NeurIPS Dataset and Benchmarks, rather ICLR. I also highly double this paper passes the bar for NeurIPS Dataset and Benchmarks. In conclusion, I think this work does not pass the bar for acceptance in ICLR. To improve the work, I think there are two approaches: (i) propose a new continual learning algorithm that can outperform the existing methods in the proposed protocol (ii) build a more realistic dataset. I will add that the latter still will make this work suitable for NeurIPS Dataset and Benchmarks.

---

### Official Review · Reviewer_gmDh · 2022-10-24

**Confidence:** 3
**Correctness:** 4
**Technical Novelty And Significance:** 2
**Empirical Novelty And Significance:** 2
**Recommendation:** 3

**Clarity, Quality, Novelty And Reproducibility:**

As I mentioned above the paper is well written and easy to follow. In my opinion the novelty is limited. The authors claim that they will open source the algorithm in case of acceptance, so I trust that it is reproducible.

**Strength And Weaknesses:**

Strengths:
- It is a well-written paper that defines the problem clearly.
- An algorithm that can generate a task-free continual learning stream properly for benchmarking is an important contribution.

Weaknesses:
- I'm not sure about the technical contribution of the paper. I understand it simulates task-free continual learning better than the existing approaches, but doesn't propose an approach to solve the problem itself.

**Summary Of The Paper:**

This paper proposes a new benchmark for task-free continual learning. They claim that there is a large gap between how task-free continual learning is defined and how it is evaluated, so they. propose an algorithm to reorder any labeled dataset into a simulated task-free continual learning stream for benchmarking.

**Summary Of The Review:**

I appreciate the proposed benchmark, but the contribution and the novelty is limited for ICLR standards.

---

### Official Review · Reviewer_pevR · 2022-10-24

**Confidence:** 4
**Correctness:** 3
**Technical Novelty And Significance:** 2
**Empirical Novelty And Significance:** 2
**Recommendation:** 5

**Clarity, Quality, Novelty And Reproducibility:**

It is challenging to follow the exact origin of the data, even though explicitly mentioned from a conceptual perspective. Some if the detail related to the hyperparameter, and data cleaning methodologies were not mentioned and it is possible to construed the detail required to implement their work.

The quality of the paper is second to none, as it is well written and a lot of attention to detail was given. However, there are instances where clarity should have been provided to understand the full context of their approach.

There is limited novelty in this paper as there are a lot of repeating conceptual inputs, framed differently within the context of this paper, however old the concepts.

**Strength And Weaknesses:**

This paper is well written, and the logic of the paper flows succinctly from one paradigm to another.

However, this paper does not outline the pragmatic pitfalls associated with the approach such as data distributions that might have the same class standard deviations, but are fundamentally different in data shape. Although these anomalies are not as common, their approach does not factor this in.

**Summary Of The Paper:**

This paper focusses on task-free continual learning from an online stream whose distribution changes continuously over time. Within the study, a few proposed methods were used to define the boundaries of the distribution in a few very elegantly written conceptual algorithms. Furthermore, they propose real-time evaluation frameworks that can visualise the impact of the class standard deviation multiplier. Finally, they demonstrate task-free continuums and the associated complexities within this.

**Summary Of The Review:**

Overall, this paper shows merit. There are a variety of interesting visualizations given, and those were explained properly within the context of the paper. There are, however, field specific visualisations that could have strengthened the argument, especially from the perspective of continuum disjoint tasks. Although not elaborated on too much, continuum disjoint tasks added a valuable addition to the strength of STF, and perhaps this contrast could have added a much needed justification to the argument.

---

### Official Review · Reviewer_ZSfX · 2022-10-25

**Confidence:** 3
**Correctness:** 2
**Technical Novelty And Significance:** 1
**Empirical Novelty And Significance:** 2
**Recommendation:** 3

**Clarity, Quality, Novelty And Reproducibility:**

The paper is clearly written and well presented. The quality can be substantially improved in the sense of the provided intuitions, the explanations and the contribution itself (please see 'Strengths and Weaknesses' section of this review). The novelty relies on the definition of task-free continual learning, but the proposed framework lacks novelty in terms of a method that would work well for this scenario - which is not proposed at all. The reproducibility is far (a researcher working in this area could eventually reproduce the results).

**Strength And Weaknesses:**

Strengths:
- The paper pursues an interesting research direction which considers continual learning without assumptions on how classes are organised into tasks. Task information is not required for learning nor for inference purposes. This is certainly a more realistic scenario of continual learning, which as the authors exemplify may be applicable to a variety of domains.
- The paper is clearly written and well presented.

Weaknesses:
- Although I appreciate the relevance of pursuing real task-free continual learning, I find the proposed method quite straightforward and not fully sound. For example, although you mention some reasons to select the Beta distribution over some other possible distributions for sampling on each class, these arguments seem quite trivial and do not necessarily imply that other distributions are not entirely applicable. For a framework such as the intended one, I would expect to see a thorough analysis of how different class distribution assignments work for the purpose of task-free continual learning. Another aspect that impacts the soundness of the proposed method is the amount of explanations on section 3.2 around maximum entropy principle and the beta distribution itself, and the lack of intuition about how class distribution assignment actually works.
- A second weakness that I see in this paper is regarding the experiments: from Figure 1, it seems that the proposed method tends to allocate examples from the same class quite contiguously; with the small batch sizes used in the experiments, it is very likely that examples from a particular class will be all observed on the same batch. I would expect to see more systematic experiments that clearly show that the task-free approach works for scenarios where examples from the same class are observed at just a few, some or many steps over the learning sequence. I would also like to see more systematically how important hyperparameters such as the batch size affect task-free continual learning.
- Finally, beyond the definitions of task-free continual learning and the reported experiments, I consider the contribution of this paper quite limited. For an ICLR paper, I would that the proposed framework would also include a method that could leverage better this task-free scenario.

**Summary Of The Paper:**

This paper presents a framework for task-free continual learning. The framework consists of a method for converting any dataset to a task-free continual learning problem where information on the class is not required and is not explicit, but rather examples from different classes can be observed at any step of the learning sequence. This method relies on assigning distributions to each class so that these distributions can be later used to select examples from a particular class, or set of classes, for a particular batch for learning. To assign these distributions, the authors propose to sample mean and standard deviations from a Beta distribution. Experiments are run over four different benchmark datasets and a range of baseline continual learning methods.

**Summary Of The Review:**

As noted in the "Strengths and Weaknesses" section, although I appreciate the definition of the task-free continual learning scenario and the attempts at experimentally measuring how different baselines work in this setting, I consider that the contribution of this paper lacks novelty and is not sufficient for ICLR since there is no method being proposed for exploiting the task-free learning scenario. Therefore, my recommendation for this paper is a rejection, but I encourage the authors to improve their contribution by incorporating a method for task-free learning as part of their framework.

---

### Decision · Program_Chairs · 2023-01-20

**Decision:**

Reject

**Justification For Why Not Higher Score:**

All reviewers recommended to reject the paper, and the authors did not provide responses to the reviews. Thus the initial concerns on the limited contributions of the work were not dealt with.

**Justification For Why Not Lower Score:**

N/A

**Metareview: Summary, Strengths And Weaknesses:**

This paper proposes a new evaluation protocol for task-free continual learning, which transforms a labeled dataset into a data stream in which the class distribution continuously change over time for instance, by forming a beta-distribution for each class over time. The authors argue that the proposed benchmark is more realistic compared to existing task-free continual learning benchmarks whose data distributions change only at discrete steps. The authors validate existing test-free continual learning methods on the proposed benchmark, and also show how their simulated task-free continual learning stream differs from existing disjoint task streams.

The reviewers in general agrees with the main argument that the proposed evaluation protocol provides a more realistic benchmark for task-free continual learning methods, and found the paper well-written. However, they are also concerned with the lack of deeper analysis of existing task-free continual learning methods, and finds the contribution of the paper rather limited since the paper does not provide a new method to tackle the new, challenging setting introduced in the paper, and unanimously leaned toward rejection.

Since the authors did not provide a rebuttal during the author-reviewer interaction period, the reviewers did not change their negative ratings on the paper. As the reviewers agree, this could be a promising research direction, but the authors should delve more deeper into the problem, if they are to write a good empirical study paper on the topic, or identify the limitations of existing methods on the new benchmark and propose a method to overcome the challenges.